# The nucleolar shell provides anchoring sites for DNA untwisting

Jumpei Fukute[1,2], Koichiro Maki [1,2,3,4✉] & Taiji Adachi [1,2,3,4]

DNA underwinding (untwisting) is a crucial step in transcriptional activation. DNA underwinding occurs between the site where torque is generated by RNA polymerase (RNAP) and the site where the axial rotation of DNA is constrained. However, what constrains DNA axial rotation in the nucleus is yet unknown. Here, we show that the anchorage to the nuclear protein condensates constrains DNA axial rotation for DNA underwinding in the nucleolus. In situ super-resolution imaging of underwound DNA reveal that underwound DNA accumulates in the nucleolus, a nuclear condensate with a core–shell structure. Specifically, underwound DNA is distributed in the nucleolar core owing to RNA polymerase I (RNAPI) activities. Furthermore, underwound DNA in the core decreases when nucleolar shell components are prevented from binding to their recognition structure, G-quadruplex (G4). Taken together, these results suggest that the nucleolar shell provides anchoring sites that constrain DNA axial rotation for RNAPI-driven DNA underwinding in the core. Our findings will contribute to understanding how nuclear protein condensates make up constraints for the site-specific regulation of DNA underwinding and transcription.

[1] Laboratory of Cellular and Molecular Biomechanics, Department of Mammalian Regulatory Network, Graduate School of Biostudies, Kyoto University, Sakyo, Kyoto, Japan. [2] Laboratory of Biomechanics, Institute for Life and Medical Sciences, Kyoto University, Sakyo, Kyoto, Japan. [3] Department of Micro Engineering, Graduate School of Engineering, Kyoto University, Sakyo, Kyoto, Japan. [4] Department of Medicine and Medical Science, Graduate School of Medicine, Kyoto University, Sakyo, Kyoto, Japan. ✉email: maki@infront.kyoto-u.ac.jp

In a cell nucleus, genomic DNA can be underwound (untwisted)[1] or overwound (twisted)[2] by mechano-biochemical factors[3], regulating the access of polymerases[4–6] and nucleases[7,8] to the genetic code. In addition, under/over-wound DNA can adopt non-B secondary structures[9,10] and supercoiled structures (plectonemes)[11], contributing to the multiscale genome configuration[12]. Thus, DNA under/over-winding plays a pivotal role in the spatiotemporal regulation of genomic processes such as transcription, replication, and repair.

According to a model proposed by Liu and Wang[13], DNA under/over-winding occurs between the site where torque is generated and the site where the axial rotation of DNA is constrained. During gene transcription, RNA polymerase (RNAP), whose rotation around the DNA helical axis is constrained, generates torque to induce axial rotation of the template DNA[14]; however, torque-generating RNAP alone is not sufficient for DNA under/over-winding[15,16]. By contrast, DNA becomes under/over-wound when DNA axial rotation is constrained behind/ahead the RNAP[15,16]. Therefore, constraints on DNA axial rotation are critical to determining the under/over-winding regions along genomic DNA.

DNA looping has been proposed as a major constraint on DNA axial rotation. DNA loop formation by bridging proteins, such as LacI in prokaryotic cells[17,18] and cohesin in eukaryotic cells[19,20], could constrain the axial rotation of DNA at the bridging sites. However, multiple underwound genomic regions (~100 kb) have been found within a single loop (>1 Mb)[21], suggesting the existence of constraints by looping-independent factors.

In this study, we hypothesized that anchoring to nuclear protein condensates[22] constrains the axial rotation of DNA. To test this hypothesis, we first identified the intranuclear location where underwound DNA is produced by performing in situ super-resolution imaging. Next, we investigated how DNA axial rotation is constrained in the nucleus by focusing on the interaction of DNA with nuclear condensates.

## Results

**Biotinylated psoralen binds to underwound DNA in the nucleus**. We fluorescently labeled underwound nuclear DNA with psoralen[19,21,23,24], an intercalator that preferentially binds to underwound DNA rather than relaxed DNA[25]. An in vitro crosslinking assay[26] verified that biotinylation of psoralen for avidin-based fluorescent labeling retains its preferential binding to underwound DNA (Supplementary Fig. 1), as previously shown by sequencing[23,27–29]. Biotinylated psoralen (bio-psoralen) was used to fluorescently label underwound DNA in MC3T3-E1 mouse osteoblast-like cells (Fig. 1a). First, cell membranes were gently permeabilized with digitonin to enhance the incorporation of bio-psoralen into the nucleus[23,30]. Then, the cells were treated with bio-psoralen and irradiated with UV light, followed by paraformaldehyde fixation, permeabilization, and fluorescent labeling with NeutrAvidin Protein, DyLight$^{TM}$ 488. To validate each staining step, we performed staining while omitting each step: digitonin treatment, bio-psoralen incorporation, UV irradiation, or NeutrAvidin Protein, DyLight$^{TM}$ 488 treatment (Fig. 1b). Only samples that had been subjected to all four staining steps (Fig. 1b, right-most panel) showed prominent nuclear signals in all cells. Blockade of the biotin–avidin interaction via treatment with free avidin diminished nuclear bio-psoralen signals (Supplementary Fig. 2a). In addition, DNA degradation by DNase I treatment decreased nuclear bio-psoralen signals (Supplementary Fig. 2b). As bio-psoralen binds to pairwise RNA secondary structures[31] and underwound DNA, we also assessed the possibility that bio-psoralen binds to nuclear RNA.

Using co-staining with StrandBrite$^{TM}$, a specific RNA marker, we observed no remarkable co-localization of bio-psoralen and StrandBrite$^{TM}$ signals (Supplementary Fig. 2c), and RNA signal was significantly decreased after RNase A treatment (Supplementary Fig. 2d, e). On the other hand, the nuclear signal of bio-psoralen was sustained (Supplementary Fig. 2d, f). These results confirmed that bio-psoralen-bound DNA were fluorescently labelled in a nucleus.

To assess whether bio-psoralen predominantly binds to DNA in the underwound state, we examined changes in its binding after treating cells with bleomycin (BLM), which induces the transition of under/over-wound DNA to relaxed state by introducing both single- and double-strand breaks[21,32]. Naughton et al. had utilized biotinylated trimethyl psoralen (bTMP, not bio-psoralen) in human cells and verified the binding specificity of bTMP by showing that BLM treatment decreased bTMP signal[21]. Consistent with Naughton's findings, BLM treatment decreased the amount of bio-psoralen-bound genomic DNA (Fig. 1c) and the fluorescence intensity of bio-psoralen in the nucleus (Fig. 1d, e). These results indicate that bio-psoralen predominantly binds to underwound DNA in the nucleus, verifying the specificity of our staining method.

**Underwound DNA colocalizes with nuclear proteins**. We identified the intranuclear location of underwound DNA and associated nuclear proteins at super-resolution using lattice-pattern structured illumination microscopy (Lattice-SIM$^2$) (Fig. 2a). Notably, bio-psoralen-labeled underwound DNA exhibited a distinct distribution from 4′,6-diamidino-2-phenylindole (DAPI), a minor grove binding dye. This difference would be explained by the stronger interaction of DAPI with overwound DNA[33], which has a decreased minor groove width[34]. Co-staining of underwound DNA and several marker proteins revealed that underwound DNA was distributed in the vicinity of RNA polymerase II phosphorylated at serine 2 (RNAPIIS2P) (Fig. 2b). This result is consistent with previous reports on the generation of underwound DNA behind RNAPII during transcription elongation[21,29]. The observation of underwound DNA in proximity with RNAPII, which was in agreement with the fact that RNAPII generates torque to induce DNA underwinding, encouraged us to further investigate using the developed methodology. Notably, several bright foci of underwound DNA existed in nuclei (Fig. 2c, arrows). In these foci, underwound DNA was distributed adjacent to clusters of topoisomerase I (TopI), which is involved in the relaxation of under/over-wound DNA[35,36], further supported that our imaging technique specifically detects underwound nuclear DNA. It is important to mention here that the distribution of TopI has been reported to span nucleoplasm[37], suggesting that some portion of proteins potentially leak to cytoplasm by digitonin treatment[38]. To assess the roles of TopI and topoisomerase II (TopII) in the regulation of DNA underwinding, we performed an inhibitor experiment using camptothecin (for TopI inhibition) and etoposide (for TopII inhibition) (Supplementary Fig. 3). Although we expected that TopI/II inhibition would increase underwound DNA and produce a stronger bio-psoralen signal, Top I/II inhibition decreased the nuclear bio-psoralen signal, indicating a decrease in the amount of underwound DNA. Such responses, as previously reported[19,21], might be caused by secondary effects of TopI/II inhibitor treatment, such as DNA strand break[36] and reduced RNAP activity[29,39]. Taken together, we successfully elucidated the intranuclear location of underwound DNA and associated nuclear proteins.

**Underwound DNA accumulates in the nucleolar core**. As TopI is known to distribute in nucleoli[37], the site of ribosome biogenesis, we examined whether DNA underwinding occurs in ribosome DNA (rDNA). We stained rDNA by fluorescence in situ hybridization (FISH) and found that bio-psoralen signal

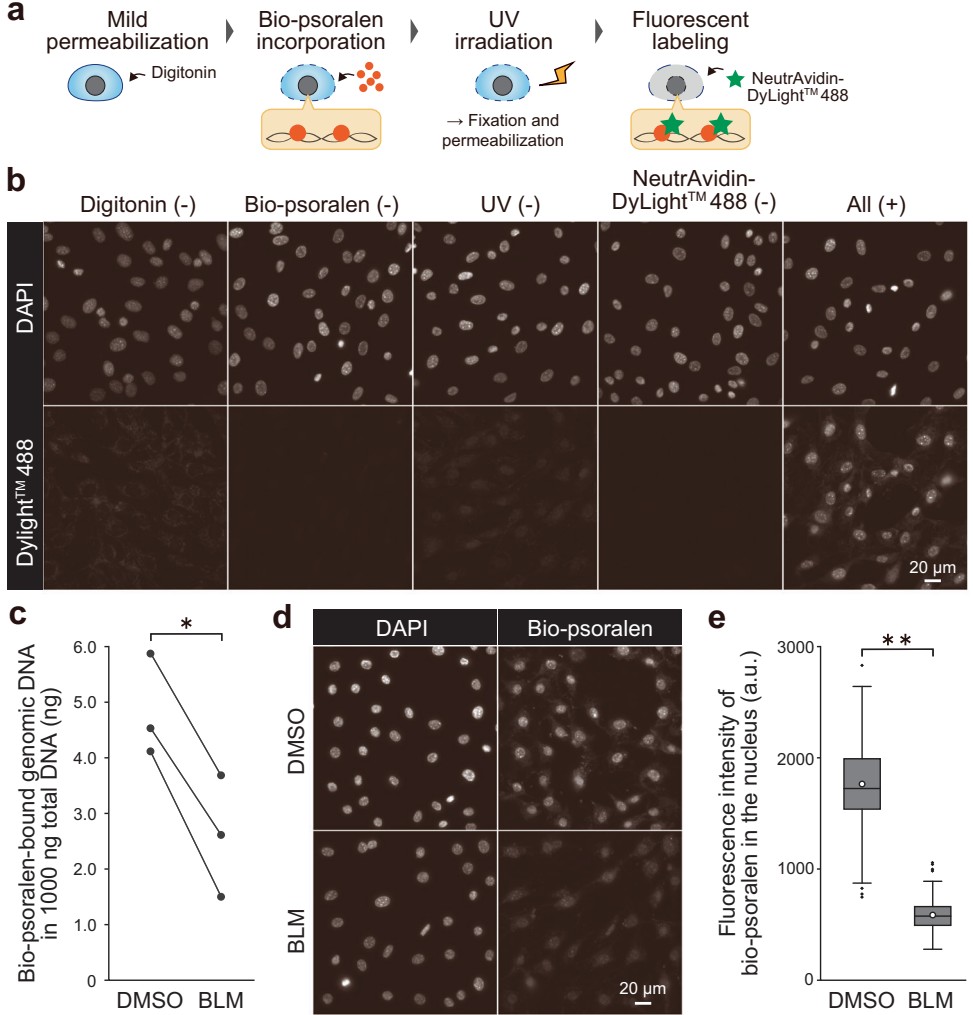

**Fig. 1 Fluorescent labeling of intracellular underwound DNA. a** Procedure for bio-psoralen fluorescent labeling. **b** Fluorescence images of DAPI controls and bio-psoralen (DyLight™ 488) in samples subjected to all or only three of the staining steps. **c** Quantification of bio-psoralen-bound genomic DNA extracted from DMSO control and BLM-treated cells (100 µM, 15 min). BLM introduces single-/double-strand breaks and induces the transition of underwound DNA to relaxed DNA. Three independent biological replicate experiments were performed ($n = 3$). Statistical significance was assessed using the two-sided paired $t$-test. $P^* < 0.05$ ($P = 0.0367$). **d** Fluorescence images of DAPI and bio-psoralen in DMSO- and BLM-treated cells. **e** Fluorescence intensity of bio-psoralen in the nucleus. The circle, centerline, box, upper/lower whiskers, and dots represent the mean, median, 75th/25th percentiles, 1.5 times the interquartile range, and outliers respectively. $n = 150$ cells. Statistical significance was assessed using the two-sided Mann–Whitney $U$-test. $P^{**} < 0.01$ ($P = 3.05 \times 10^{-50}$).

partly overlapped with that of rDNA (Fig. 3a). This result indicated that DNA underwinding occurs in rDNA as well.

Nucleoli have a core–shell structure consisting of three sub-compartments[40]: fibrillar center (FC) and dense fibrillar component (DFC) forming the core, and granular component (GC) forming the shell (Fig. 3b). This structure enables the successive steps of ribosome biogenesis, with rRNA transcription taking place at the FC/DFC interface, processing in the DFC, and assembly with ribosomal proteins in the GC[40,41]. Although digitonin potentially affects the integrity of the nucleolar sub-compartments[38], we confirmed that the characteristic nucleolar structure where shell component proteins surround core component proteins was retained after digitonin treatment (50 µg/mL, 1 min) (Supplementary Fig. 4). Under this condition of digitonin treatment, we co-stained bio-psoralen and marker proteins of each sub-compartment, including RNA polymerase I (RNAPI, an FC marker), fibrillarin (FBL, a DFC marker), and nucleophosmin (NPM, a GC marker). As a result, we found that underwound DNA was mainly distributed in the FC and DFC

(Fig. 3c), indicating that underwound DNA is generated in the vicinity of RNAPI in the FC and extends to the DFC. Compared to the region inside the nucleolus, underwound DNA was less abundant outside the nucleolus (Fig. 3d, e). These results suggested that DNA axial rotation is constrained in the GC (shell), resulting in RNAPI-driven DNA underwinding in the FC and DFC (core).

**DNA axial rotation is constrained by anchoring to the nucleolar shell via GC protein–G-quadruplex (G4) interaction.** To identify the driving force of DNA underwinding, we performed inhibitor experiment for RNAPI. The prominent bio-psoralen foci in nucleoli disappeared after inhibition of RNAPI transcription initiation using CX-5461[42] as well as inhibition of elongation using Actinomycin D (ActD) (Fig. 4a). The intensity profile also showed that the bio-psoralen signal was significantly decreased inside the nucleolus (Fig. 4b–d). Of note, in these experiments, the treatment time was limited to 15 min, to avoid

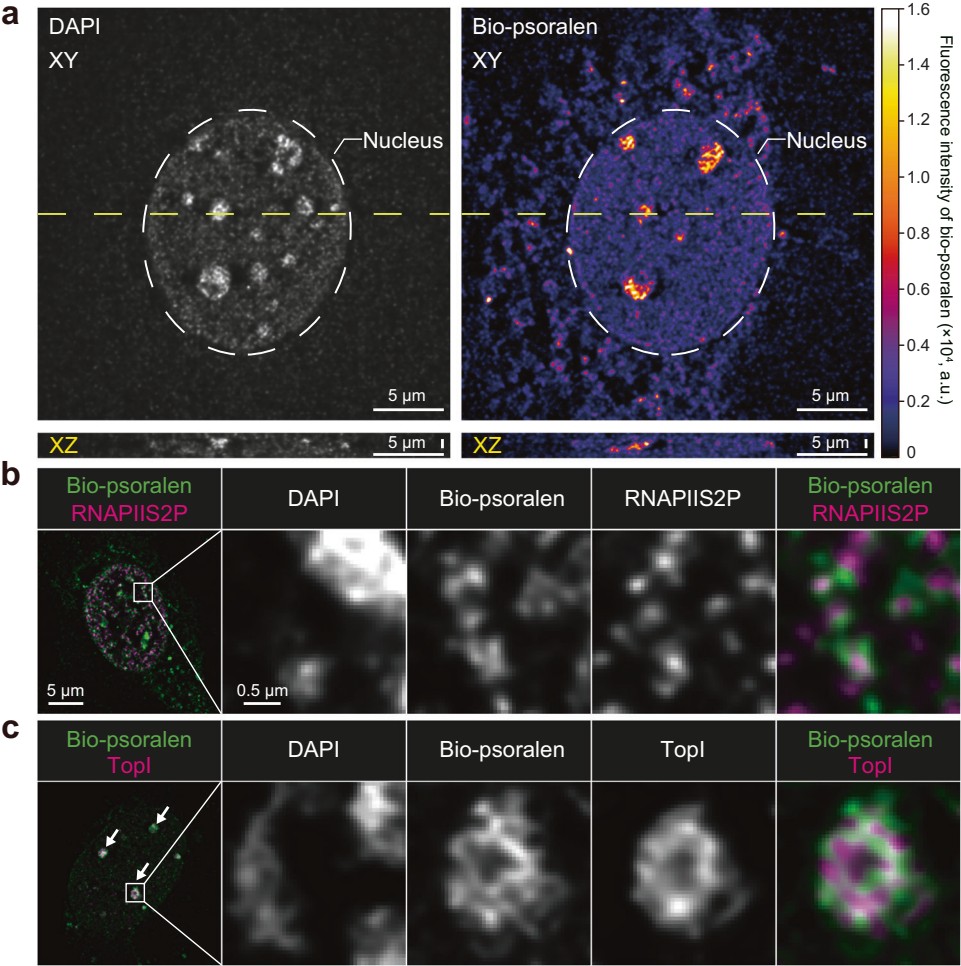

**Fig. 2 In situ super-resolution imaging of underwound DNA. a** Co-staining of DAPI (left) and bio-psoralen (right) displayed in XY (top) and XZ (bottom) sections along the dashed yellow line in the XY section. Heatmap is shown for bio-psoralen fluorescence intensity. **b** Co-staining of DNA (DAPI), bio-psoralen, and RNAPIIS2P. **c** Co-staining of DNA (DAPI), bio-psoralen, and TopI. Arrows indicate prominent foci of bio-psoralen.

cell death (Supplementary Fig. 5). These results indicated that DNA is underwound by RNAPI transcription in nucleoli.

Having identified the driving force of DNA underwinding in the nucleolus, we next explored possible factors that constrain DNA axial rotation. We focused on DNA anchoring to the GC as a factor constraining DNA axial rotation. The GC is a condensate that consists of rRNA and self-assembling proteins, including NPM[40]. NPM is known to interact with the G-quadruplex (G4) formed in rDNA[43], a non-B DNA structure composed of stacked G-tetrads[44]. Based on these reports, we examined whether DNA axial rotation is constrained by DNA anchoring to the GC via the interaction between NPM and G4. We performed an inhibition experiment using the G4-selective ligand TMPyP4, which inhibits the interaction with NPM[43]. As we expected, chromatin immunoprecipitation (ChIP) analysis showed that TMPyP4 treatment prevented NPM from binding to putative G4 forming sequences (PQS) in rDNA (Fig. 5a). Then, cells treated with or without TMPyP4 were stained with NPM and bio-psoralen (Fig. 5b). A lower NPM signal was detected in TMPyP4 treated cells because NPM could not bind to G4, and subsequently left the nucleoli and diffused into the nucleoplasm[43]. Importantly, bio-psoralen signal in the nucleolus was significantly decreased by TMPyP4 treatment (Fig. 5c, d). rDNA still existed in the nucleolus after TMPyP4 treatment (Fig. 5e), indicating that the inhibition of NPM-G4 interaction relaxes the underwound rDNA inside the nucleolus. This result suggested that the release of

NPM from G4 by TMPyP4 treatment allowed DNA axial rotation, resulting in the transition of underwound DNA to relaxed DNA. These findings suggest that DNA axial rotation is constrained by anchoring to the nucleolar shell via GC protein–G4 interaction, contributing to RNAPI-driven DNA underwinding in the core (Fig. 5f).

## Discussion

The in situ imaging analysis of underwound DNA demonstrated that anchoring to nuclear condensates constrains the axial rotation of DNA, facilitating RNAP-driven DNA underwinding. We showed that nucleoli contain torque-generating RNAP in their cores and anchoring sites in their shells, enabling the accumulation of underwound DNA. In the nucleolar shell (GC), proteins with self-assembling and DNA-binding domains may contribute to the constraint of DNA axial rotation. NPM, a major protein in the GC, forms pentamers via its N-terminal domain[45] and assembles with nucleolar proteins, rRNA, and itself via an intrinsically disordered region to form huge condensates[46,47] that bind to rDNA via G4-binding C-terminal domains[43,48]. This exerts high rotational drag to constrain DNA axial rotation at the NPM-bound region. This mechanism can be generally applied to RNAPII-mediated transcription, in which DNA-binding transcription factors assemble with coactivators to form condensates[49]. The assembly of transcription factors is

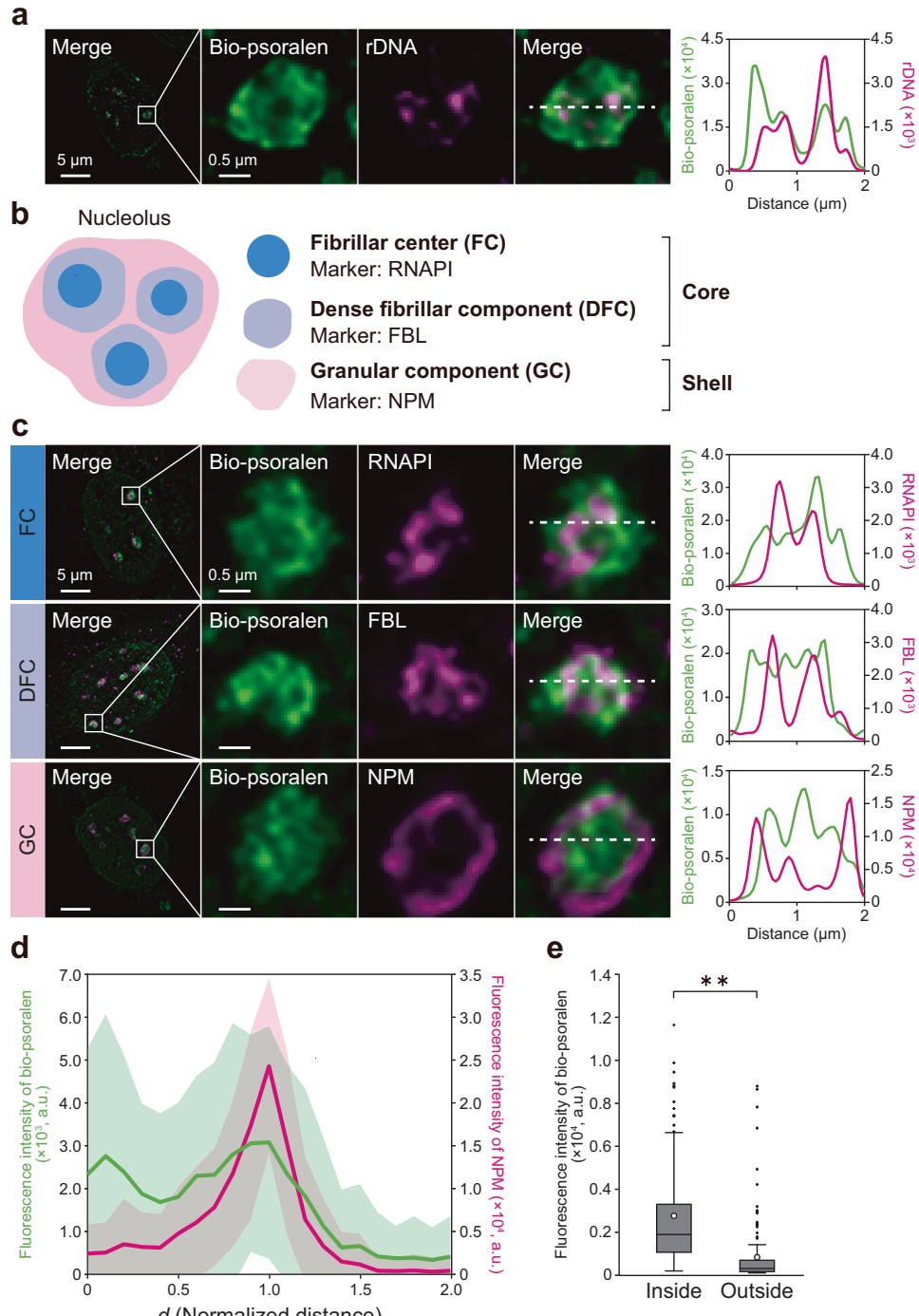

**Fig. 3 Accumulation of underwound DNA in the nucleolar core. a** Co-staining of bio-psoralen and ribosome DNA (rDNA). **b** Diagram of nucleolar sub-compartments; FC and DFC form the core and GC forms the shell. **c** Co-staining of bio-psoralen and RNAPI (top), fibrillarin (middle), and NPM (bottom). **d** Intensity profile of bio-psoralen and NPM. Intensity profile in each nucleolus was measured along the radial axis from the centroid of the nucleolar outline, which was identified with NPM. The normalized distance $d$ was calculated based on the distance between the centroid and the intersection of radial axis and the nucleolar outline. $n = 68$ nucleoli/11 cells. The intensity profile was smoothed by moving-average. The standard deviation is shown as a band. **e** Mean intensity of bio-psoralen inside ($0 \leq d \leq 0.5$) and outside ($1.5 \leq d \leq 2.0$) the nucleolus. $n = 68$ nucleoli/11 cells. Statistical significance was assessed using the two-sided Mann–Whitney $U$-test. $P^{**} < 0.01$ ($P = 1.58 \times 10^{-25}$).

dynamically regulated by phosphorylation[50] and ligand binding[51]. These dynamic processes alter the degree to which DNA axial rotation is constrained and possibly contribute to the spatiotemporal regulation of DNA underwinding.

We found that G4 is a key DNA structure for anchoring to the GC. As G4 is enriched in promoter regions[44], underwound DNA

is expected to accumulate in promoter regions of rDNA tandem repeats. Furthermore, DNA underwinding promotes G4 formation by facilitating DNA strand separation[9,10]. Taken together with our findings, this information suggests a positive feedback mechanism of DNA underwinding where G4-mediated constraints facilitate DNA underwinding, which, in turn, promotes

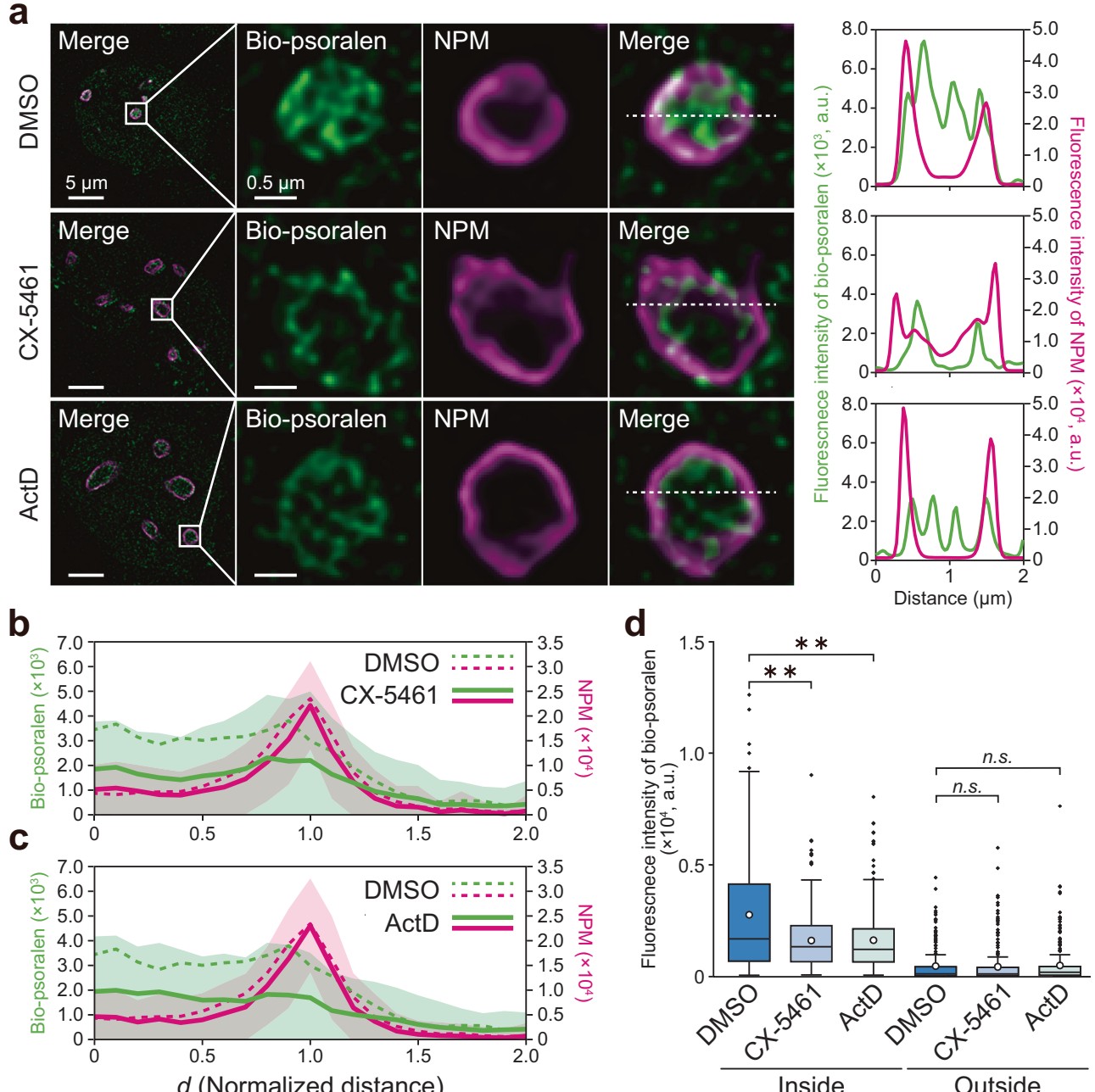

**Fig. 4 DNA underwinding in the nucleolus is driven by RNAP I transcription. a** Co-staining of bio-psoralen and NPM in DMSO-, 3 μM CX-5461- and 50 nM Actinomycin D (ActD)-treated cells. The treatment time is 15 min. CX-5461 and Act. D inhibit the RNAPI activities in the initiation and elongation steps, respectively. **b** Intensity profile of bio-psoralen and NPM in CX-5461-treated cells ($n = 135$ nucleoli/19 cells). The dotted line shows the intensity profile in DMSO-treated cells ($n = 89$ nucleoli/13 cells). **c** Intensity profile of bio-psoralen and NPM in ActD-treated cells ($n = 120$ nucleoli/17 cells). **d** Mean intensity of bio-psoralen inside ($0 \leq d \leq 0.5$) and outside ($1.5 \leq d \leq 2.0$) the nucleolus. $n = 89$ nucleoli/13 cells for DMSO-treated cells, $n = 135$ nucleoli/19 cells for CX-5461-treated cells and $n = 120$ nucleoli/17 cells for ActD-treated cells. Statistical significance was assessed using the two-sided Steel-Dwass test. $P^{**} < 0.01$ ($P = 0.00118$ for DMSO vs CX-5461, $P = 0.001$ for DMSO vs ActD). *n.s.*, not significant ($P = 0.9$ for DMSO vs CX-5461, $P = 0.353$ for DMSO vs ActD).

G4 formation via DNA strand separation. As G4 is prevalent in genomic DNA[44], G4-mediated positive feedback may play a pivotal role in regulating DNA underwinding.

It should be noted that factors other than anchoring to the nucleolar shell could constrain DNA axis rotation in nucleoli. For example, upstream binding factor (UBF) binds to rRNA coding region while forming a nucleoprotein structure called enhancesome[52]. DNA looping by enhancesome formation might constrain DNA axial rotation, in a similar way to that of LacI[17,18].

G4 formation itself has also been suggested to constrain DNA axis rotation. In vitro experiments previously showed that G4 formation hinders the transmission of negative supercoil (underwound DNA) due to the formation of a more rigid structure[53]. Besides anchoring to the nucleolar shell, these factors could be constraints and determine the genomic landscape of under/overwound DNA.

In this study, we identified RNAPI transcription as the driving force of DNA underwinding in the nucleolus. RNAPI transcribes

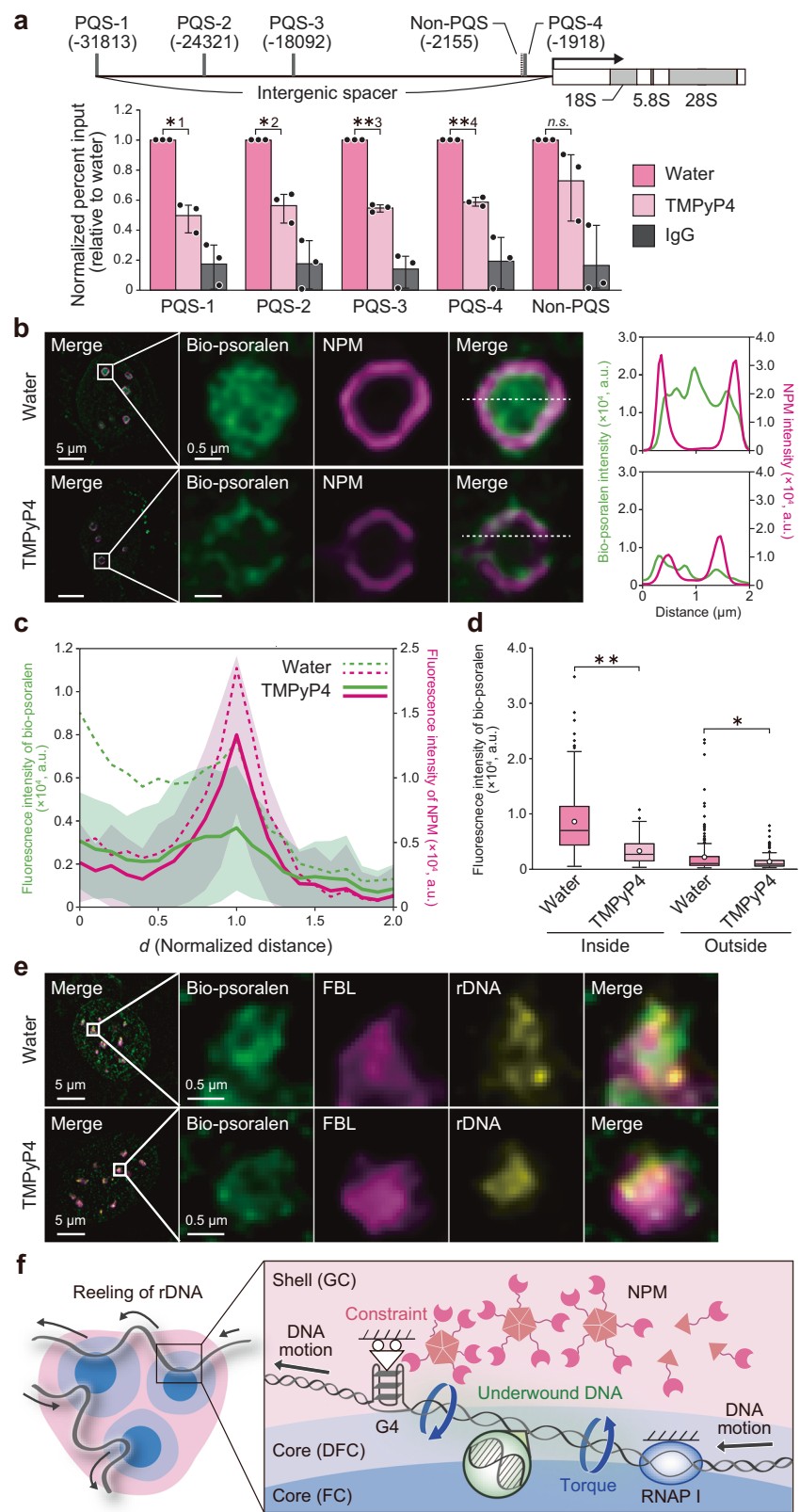

faster and pauses less often than RNAPII owing to the intrinsic backtracking recovery activities[54], suggesting that RNAPI exhibits stronger torque capacity than RNAPII[55]. This efficient torque generation by RNAPI would help the accumulation of underwound DNA in the nucleolus. In the future, single molecule experiments such as direct torque measurements[14,56], would reveal how RNAPI generates torque and how it differs from RNAPII/III. Besides the type of RNAP, transcriptional activities in each genomic region also affect DNA underwinding. Mouse rDNA consists of more than 100 tandem repeats and 30–50% of them are transcriptionally active[57]. The variation in transcriptional activities of each repeat would generate the non-uniform

**Fig. 5 GC protein–G4 interaction locally constrains axial rotation to induce nucleolar DNA underwinding. a** qPCR analysis for chromatin immunoprecipitation (ChIP) DNA samples using NPM antibody for cells without (water) and with TMPyP4 treatment (100 μM, 15 min). TMPyP4 is a G4-selective ligand that inhibits the interaction with NPM. For the negative control, ChIP samples with normal rabbit IgG was used. Each target region for qPCR is shown in the upper panel with the relative position from the transcription start site. Four putative quadruplex sequences (PQS) and one non-PQS were targeted. The percent input (%) was normalized by the control (water) sample. Three independent biological replicate experiments were performed ($n = 3$). Statistical significance was assessed using the one-sample $t$-test. $P^{*1} < 0.05$ ($P = 0.0132$), $P^{*2} < 0.05$ ($P = 0.0177$), $P^{**3} < 0.01$ ($P = 0.000997$), $P^{**4} < 0.05$ ($P = 0.00175$). n.s., not significant ($P = 0.185$). All bar graphs show mean ± s.d. **b** Fluorescence images of bio-psoralen and NPM in cells without (water) and with TMPyP4 treatment (100 μM, 15 min). The right panel is a line profile of the fluorescence intensity of bio-psoralen and NPM along the dashed lines in the merged images. **c** Intensity profile of bio-psoralen and NPM in TMPyP4-treated cells ($n = 84$ nucleoli/16 cells). The dotted line shows the intensity profile in water-treated cells ($n = 108$ nucleoli/15 cells). **d** Mean intensity of bio-psoralen inside ($0 \leq d \leq 0.5$) and outside ($1.5 \leq d \leq 2$) the nucleolus. $n = 108$ nucleoli/15 cells for water-treated cells and $n = 84$ nucleoli/16 cells for TMPyP4-treated cells. Statistical significance was assessed using the two-sided Mann–Whitney $U$-test. $P^{**} < 0.01$ ($P = 3.35 \times 10^{-26}$), $P^{*} < 0.05$ ($P = 0.0148$). **e** Immuno-FISH staining for bio-psoralen, FBL and rDNA for water-treated cells and TMPyP4-treated cells (100 μM, 15 min). **f** Proposed mechanism of DNA underwinding in nucleoli. During transcription, rDNA is reeled through RNAPI condensates (FC/DFC interface). Anchoring of the rDNA to the GC (shell) via NPM–G4 interaction constrains the axial rotation of DNA, facilitating DNA underwinding between RNAPI and the anchoring sites. This leads to the accumulation of underwound DNA in the FC and DFC (core).

distribution of underwound DNA along rDNA, as shown in Fig. 3a.

This study has potential limitations. First, the bio-psoralen fluorescence imaging method used in this study involves digitonin treatment, which potentially causes the leakage of DNA and soluble proteins from the cell nucleus[38]. An attenuation in the nucleoplasmic distribution of TopI (Fig. 2c) and NPM (Fig. 3c) in digitonin-treated cells could be caused by the leakage of the soluble fraction of each protein. Second, the bio-psoralen-based approach cannot directly detect overwound DNA. Therefore, it remains unknown whether overwound DNA also accumulates in the nucleolus, and if so, how DNA overwinding occurs. It is necessary to develop new probes for overwound DNA by utilizing its binding protein such as GapR, which is helpful for genome-wide mapping of overwound DNA[2].

DNA underwinding facilitates RNAP recruitment to promoter regions[4,5], whereas excess DNA underwinding is known to promote the release of RNAP from the promoter region[30]. In the nucleus, under/over-wound DNA is relaxed by topoisomerases[21]. In the present study, TopI was found adjacent to underwound DNA foci in nucleoli (Fig. 2c). Topoisomerase II (TopII), which relaxes supercoiled DNA[35], is also predominantly localized in nucleoli[58]. Therefore, nucleolar RNAPI-mediated generation of underwound DNA could be coupled to TopI/II-mediated relaxation, enabling subsequent rRNA transcription. The regulatory mechanisms of underwound DNA generation and relaxation need to be studied using simultaneous live imaging of underwound DNA, RNAP, and topoisomerases.

## Methods

**In vitro biotinylated (bio-)psoralen crosslinking assay.** The intercalation of bio-psoralen into underwound DNA was examined using an in vitro crosslinking assay[26]. DNA was crosslinked by bio-psoralen under UV irradiation, linearized, and denatured by heat. As bio-psoralen crosslinking prevents DNA heat denaturation, the intercalation of bio-psoralen can be evaluated based on the amount of non-denatured DNA after heat treatment. Relaxed and underwound DNA were prepared by incubating plasmid pBR322 (Takara Bio) with topoisomerase I (TopI, Takara Bio) and gyrase (TopoGEN), respectively, at 37 °C for 1 h, and purified using NucleoSpin® Gel and a PCR Clean-up kit (Macherey-Nagel). Bio-psoralen (200 μM; EZ-Link® Psoralen-PEG$_3$-Biotin, Thermo Fisher Scientific) was added to DNA and exposed to 3.6 kJ/m$^2$ of long-wave (365 nm) UV (LUV-4; AS ONE) on ice for 30 min. For linearization, DNA was cleaved with *EcoR*I (Toyobo) at 37 °C for 1 h. For heat denaturation, one half of each sample was boiled for 5 min and immediately chilled in

ice-cold water. Agarose gel electrophoresis was run at 50 V for 1 h, and the gel was stained with 0.1 μg/mL ethidium bromide solution (Nacalai Tesque) for 30 min. Images were acquired using ImageQuant™ LAS500 (Cytiva) and DNA bands were quantified using Fiji[59]. The fraction of bio-psoralen-crosslinked DNA was calculated by dividing the amount of non-denatured DNA by the total amount of DNA.

**Cell culture.** MC3T3-E1 mouse osteoblast-like cells (ATCC) were cultured in minimum essential medium α (Gibco) containing 10% fetal bovine serum (Gibco) and 1% antibiotic-antimycotic (Nacalai Tesque) in a humidified incubator at 37 °C under a 5% $CO_2$ atmosphere. For fluorescence imaging, $1 \times 10^4$ cells were seeded on a 7-mm-diameter glass bottom dish (MatTek) for 24 h prior to imaging. For genomic DNA extraction, $2 \times 10^6$ cells were cultured on a 100-mm-diameter Nunc™ EasYDish (Thermo Fisher Scientific). For chromatin immunoprecipitation, $4 \times 10^6$ cells were cultured on a 150-mm-diameter cell culture dish (Corning).

To verify the specificity of bio-psoralen for underwound DNA, a perturbation experiment was conducted. Cells were treated with 100 μM bleomycin (Cayman Chemical), which introduces DNA strand breaks and induces the transition of underwound DNA to relaxed DNA[21,32], in complete medium at 37 °C for 15 min. To examine the roles of TopI/II in the regulation of DNA underwinding, cells were treated with 10 μM camptothecin (for TopI inhibition) and 10 μM etoposide (for TopII inhibition) in complete medium at 37 °C for 15 min. To inhibit RNAPI transcription initiation and elongation, cells were treated with 3 μM CX-5461 (ChemScene LLC)[42] and 50 nM Actinomycin D (Nacalai tesque), respectively, in complete medium at 37 °C for 15 min. To inhibit the interaction between nucleolar proteins and G-quadruplex (G4), cells were treated with 100 μM TMPyP4 (Abcam)[43], a selective G4 ligand, in complete medium at 37 °C for 15 min.

**In situ fluorescence staining of underwound DNA in the cell nucleus.** Fluorescence staining using bio-psoralen was conducted according to previous reports[19,21,23,24], with modification. To avoid cell detachment from the growth dishes, digitonin and bio-psoralen were dissolved in phosphate-buffered saline (PBS) (+), which was prepared by adding calcium and magnesium solution (Nacalai Tesque) to PBS (−) (Gibco) at a 1:100 dilution. Cells were treated with 50 μg/mL digitonin (Cayman Chemical) in PBS (+) at 37 °C for 1 min in the dark. After rinsing with PBS (+), 20 μM bio-psoralen in PBS (+) was added and the cells were further incubated at 37 °C for 20 min in the dark. Then, the cells

were exposed to 3.6 kJ/m$^2$ of 365 nm UV on ice for 30 min to crosslink bio-psoralen with DNA. The cells were rinsed three times with PBS (+) and fixed in 4% paraformaldehyde (PFA) in PBS (−) (Nacalai Tesque) for 15 min. The cells were permeabilized with 0.5% TritonX-100 (MP Biomedicals) in PBS (−) for 30 min, followed by blocking with 1% bovine serum albumin (BSA; Sigma-Aldrich) in PBS (−) for 30 min. After rinsing with PBS (−), bio-psoralen was fluorescently labeled by incubating the cells with 2.5 μg/mL NeutrAvidin Protein, DyLight$^{TM}$ 488 or NeutrAvidin Protein, DyLight$^{TM}$ 650 (Thermo Fisher Scientific) in 1% BSA, 0.03% Triton X-100 in PBS (−) at 4 °C overnight. For bio-psoralen staining in RNA- and DNA-degraded cells, cells were fixed by methanol for 5 min, permeabilized by 0.5% TritonX-100 in PBS (−) for 15 min, then treated with 1 mg/mL RNase A (Nippon Gene) or 50 U/mL DNase I (Takara) at 37 °C for 1 h. After RNA staining was performed as described below, bio-psoralen was labeled with NeutrAvidin Protein, DyLight$^{TM}$ 650. For bio-psoralen staining upon inhibition of biotin–avidin interaction, cells were incubated with free avidin (Avidin/Biotin Blocking Kit, Vector Laboratories) at room temperature for 1 h prior to incubation with NeutrAvidin Protein, DyLight$^{TM}$ 488.

**Immunofluorescence and chemical staining**. For immunofluorescence staining of nuclear proteins, cells were fixed, permeabilized, and blocked with BSA as described above. The cells were incubated with the following primary antibodies in 1% BSA and 0.03% Triton X-100 in PBS (−) at 4 °C overnight: anti-RNA polymerase II CTD repeat YSPTSPS (phospho S2) (5095, Abcam; 1:500), anti-topoisomerase I (85038, Abcam; 1:500), anti-RPA194 monoclonal (48385, Santa Cruz Biotechnology; 1:100), anti-fibrillarin monoclonal (2639, Cell Signaling Technology; 1:500), and anti-NPM 1 monoclonal (32–5200, Invitrogen; 1:500) antibodies. After washing with PBS (−), the cells were incubated with the following secondary antibodies in 1% BSA and 0.03% Triton X-100 in PBS (−) at room temperature for 1 h: Goat Anti-Rabbit IgG H&L-Alexa Fluor® 568 (175471, Abcam; 1:500) and Goat Anti-Mouse IgG H&L-Alexa Fluor® 568 (175473, Abcam; 1:500). For nuclear staining, cells were incubated with 4′,6-diamidino-2-phenylindole (DAPI; Invitrogen) at 1:500 at room temperature for 1 h. For RNA staining, cells were incubated with StrandBrite$^{TM}$ (AAT Bioquest) at 1:4000 at room temperature for 30 min. For cell viability assays, cells were incubated with 1 μM SYTOX Orange$^{TM}$ (Thermo Fisher Scientific) and 20 μg/mL bisBenzimide H 33342 trihydrochloride (Hoechst33342; Sigma-Aldrich) in complete medium at 37 °C for 30 min prior to the observation.

**Immuno-FISH**. For the simultaneous imaging of underwound DNA, nucleolar proteins, and ribosome DNA, Immuno-FISH was performed as previously described[60]. In brief, bio-psoralen and nucleolar proteins (fibrillarin) were fluorescently stained as described above, except that PFA fixation time was changed from 15 min to 20 min. Then, cells were again treated with 4% PFA for 10 min to crosslink antibodies. After washing with 0.05% TritonX-100 in PBS (−) three times, cells were treated with 1 mg/mL RNase A at 37 °C for 30 min. Then cells were treated with 0.1 N HCl at room temperature for 2 min, re-fixed with 1% PFA for 5 min each, and dehydrated using 70% and 100% ethanol for 5 min. Cells were air-dried for 5 min, and incubated with Cy3-labeld probes for mouse ribosome DNA (BAC clone RP23-225M6, Chromosome Science Laboratory) at 40 °C for 2 h. After heat denaturation at 70 °C for 5 min, cells were incubated at 37 °C overnight and washed with 0.01% TritonX-100 in 2× SSC buffer (Nacalai tesque).

**Microscopic observation and image analysis**. Stained cells were observed using an epifluorescence microscope (IX-83, EVIDENT) and a CCD camera. Lattice-pattern structured illumination microscopy (Lattice-SIM$^2$, Zeiss) observation was performed at room temperature using an Elyra7 super-resolution microscope (Zeiss) equipped with a Plan-Apochromat 40×/1.4 oil objective lens, four diode laser beams (405, 488, 561, and 642 nm), and an sCMOS camera, using sequential scanning of frames with a step size of 0.2–0.4 μm. Images were analyzed using Fiji[59]. To quantify the fluorescence intensity of bio-psoralen in the nucleus, nuclear outlines were generated by thresholding for DAPI staining, and mean fluorescence intensities were quantified within nuclear outlines. In addition, 3D suite[61] was utilized to identify the centroid of the nucleolus.

**Quantification of bio-psoralen-bound DNA extracted from the cell nucleus**. Bio-psoralen-bound DNA was extracted from the cell nucleus and quantified. After bio-psoralen incorporation as described above, genomic DNA was extracted by ethanol precipitation using a NucleoSpin® Tissue kit (Macherey-Nagel) and dissolved in 100 μL of Tris-EDTA buffer (pH 8.0). After treatment with 1 mg/mL RNase A (Nippon Gene) for 1 h, the genomic DNA was fragmented by applying 30 cycles of 30 sec sonication and 30 sec pauses using Bioruptor® II (BM Equipment) in ice-cold water. For biotin–avidin interaction-based purification, 100 μL Magnosphere$^{TM}$ MS300/Streptavidin (JSR Life Sciences LLC) was washed according to the manufacturer's instructions. Fragmented DNA was mixed with 2× binding buffer (20 mM Tris-HCl, 1 mM EDTA, 2 M NaCl, and 0.1% Tween 20), added to the magnetic beads, and rotated at room temperature in the dark for 1 h, followed by overnight rotation incubation at 4 °C in the dark. Unbound DNA was removed by washing the magnetic beads according the manufacturer's instructions. To release DNA from the magnetic beads, the samples were incubated in 50 μL formamide with 10 mM EDTA at 95 °C for 10 min. Finally, single-stranded DNA was quantified using Qubit$^{TM}$ (Invitrogen).

**Chromatin immunoprecipitation (ChIP) and quantitative PCR (qPCR)**. For analyzing the interaction between NPM and G4, ChIP-qPCR was performed. ChIP samples were prepared using simple SimpleChIP Enzymatic Chromatin IP Kit (Cell Signaling), according to the manufacturer's instructions. Briefly, cells were treated with formaldehyde to crosslink proteins to DNA, followed by nuclei extraction. Nuclei were treated with 0.25 μL of Micrococcal Nuclease (2000 gel units/μL) and sonicated using Bioruptor® II (BM Equipment) in ice-cold water by applying three cycles of 30 sec sonication with 30 sec pauses between each cycle. Then, 2% volume of digested chromatin was transferred to another tube for input samples. The remaining chromatin was incubated at 4 °C overnight either with anti-NPM 1 monoclonal (32–5200, Invitrogen; 10 μg), anti-Histone H3 monoclonal (4620, Cell signaling; 10 μg), and Normal Rabbit IgG (2729, Cell Signaling; 10 μg). Antibody-bound chromatin fragments were collected using Protein G Magnetic Beads. All samples including input samples were treated with NaCl and Proteinase K to revere cross-links, followed by DNA purification. Real-time PCR was performed using TB Green Premix Ex Taq GC (Takara) on a StepOne Real-Time PCR System (Thermo Fisher Scientific). Putative G4 forming sequences were predicted by G4 hunter web[62] and were used for primer design for qPCR. Primer sequences are shown in Table 1.

**Statistics and reproducibility**. All experiments performed in the manuscript were repeated at least two times as independent experiments/biological replicates. No statistical methods were

**Table 1 Primer sequences for ChIP-qPCR targeting ribosome DNA.**

| Target region | Forward | Reverse |
|---|---|---|
| PQS-1 | ACCAGTACTCCGGGCGACACTT | AAAAGAGTCCAGAGTGAGCCCGC |
| PQS-2 | AGGAACATTTGCAGTCAGTCAGT | TCTGCCTCCGAAGTGCTGGGAT |
| PQS-3 | TGGACTGACTGGCTGCCTTCCT | TGGTGCCCTCGTCTAGTGTGTC |
| PQS-4 | TGGCCAAAGCAGACCGAGTTGC | TGACGGGGACAAGAGAGGGCTT |
| Non-PQS | AACGCTCCAGGCCTCTCAGGTT | ACGACACCATCTCCGAGACGCT |

used to predetermine the sample sizes. No datapoints were excluded from the analyses. Statistical analyses were performed using SciPy[63] software. Statistical significance was determined by the specific tests indicated in the corresponding figure legends. Two-sided paired $t$-test was performed for the comparison of the amount of bio-psoralen-bound genomic DNA. Two-sided Mann–Whitney $U$-test was performed for the comparison of fluorescence intensity between two groups. Two-sided Steel-Dwass test was performed for the comparison of fluorescence intensity between three groups. One sample $t$-test was performed for the normalized percent input in ChIP-qPCR with the expected value of 1 for the null hypothesis.

**Reporting summary**. Further information on research design is available in the Nature Portfolio Reporting Summary linked to this article.

## Data availability
The datasets generated and/or analyzed during the current study are available from the corresponding author on reasonable request. The source data behind the graphs in the paper can be found in Supplementary Data 1.

## Code availability
All codes were performed by publicly available functions of Fiji (ImageJ, version 1.53t) and SciPy (version 1.11.1). No custom functions were written for the analysis.

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

## Acknowledgements

This work was supported by JST, CREST, Grant Number JPMJCR2023, Japan (J.F., K.M.), JST, CREST, Grant Number JPMJCR22L5, Japan (T.A.), JST, SPRING, Grant Number JPMJSP2110, Japan (J.F.), MEXT/JSPS, KAKENHI, Grant Number JP23KJ1255, Japan (J.F.), and MEXT/JSPS, KAKENHI, Grant Number JP20K20180 and JP23K17196, Japan (K.M.). We thank Dr. Fumiyoshi Ishidate in ZEISS-iCeMS Innovation Core Imaging Unit, Kyoto University, Japan, for supporting Lattice-SIM[2] imaging and discussions.

## Author contributions

Conceptualization: J.F., K.M. and T.A.; Methodology: J.F. and K.M.; Investigation: J.F.; Visualization: J.F., K.M. and T.A.; Funding acquisition: J.F., K.M. and T.A.; Project administration: K.M.; Supervision: T.A.; Writing—original draft: J.F.; Writing—review & editing: K.M. and T.A.

## Competing interests

The authors declare no competing interests.
