## [Peer Review File · Communications Biology]

Reviewers' comments:

Reviewer #1 (Remarks to the Author):

Fukute et al. performed some experiments to study the function of the nucleolus in the constrain of DNA axial rotation for DNA underwinding. The data are potentially interesting but very preliminary. Some suggestions follow:

- The authors should mention the existency of overwound DNA in the introduction and results. Making just a distinction between underwound and relaxed DNA this can bias the interpretation of the results.
- Section 2.1: "Biotinylated psoralen binds to underwound DNA in the nucleus" this has been previously shown by numerous publications cited in the manuscript and others. Just to make an example ref 22 (Naughton et al NSMB 2013) has mapped the presence of underwound and overwound regions in human cells and have stained nuclei with bTMP showing how bleomycin treatment decreases bTMP signal. As this is previous knowledge it cannot be presented as original findings.
- Supplementary figure 1 is quite unclear. What are we supposed to see in the agarose gel between the different samples and in heated/not heated samples? Is the signal measured in the plot in C the agarose band intensity? Also, to be able to compare signals between control, TopI and Gyrase conditions (as done in S1C) it would be appropriate to show all bands from a single gel. Can the authors provide this?
- The control carried out to exclude binding of psoralen to RNA (Supp Fig 2B) is not clear. It would be better to perform a quantification to exclude co-localization. Psoralen seems to be localized in spots in the nucleus, can you explain why it has this type of distribution?
- Figure 2. The authors detect a large amount of Bio-psoralen outside the nucleus. This is quite surprising. They conclude that this is due underwound DNA in mitochondria claiming colocalization with TFAM. It is not clear how much of the signal colocalize with mitochondrial markers as the images of bio-psoralen and TFAM do not show a very clear colocalization pattern and a quantification of the extent of colocalization is not provided. Also, the picture shown (2B, it would be nice to see more images) displays a very large amount of extranuclear signal, even more intense than the nuclear signal. It is important to ensure that the process of permeabilization with digitonin/Bio-psoralen incorporation/UV irradiation has not created some leakage of nuclear DNA. Moreover, in 2C DAPI and psoralen signal exclude each other. Does DAPI not stain underwound DNA? Also, authors say "DAPI stained DNA in a relaxed state" disregarding the existence of overwound DNA. In Fig 2D the authors refer to bio-psoralen positive regions as underwound DNA but the presence of DNA marked by DAPI is not clear in these regions. In Fig 2E, according to normal staining of TopI one would expect TopI signal a more extended area of the nucleus and not exclusively in few bio-psoralen foci. How do the authors explain this?
- The authors stated "we minimized the digitonin treatment time to 1 min and confirmed that the core-shell structure was conserved after bio-psoralen imaging." There are no results included for this important control. Indeed, as I mentioned above digitonin treatment might cause trivial leakage of nuclear DNA in the cytoplasm.
- Part of section 2.3 is based on "as RNAPI inhibition simultaneously disrupts the nucleolar structure that likely is necessary for DNA underwinding" implying than the nucleolar proteins rather than transcription itself are needed for underwinding. This is a bit misleading as it might imply that RNAPI is dispensable for underwinding or not the main driver, authors should consider rephrasing this.
- The authors show that underwound DNA is generated in nucleolar FC and DFC. This is interesting, however they also stated that underwound DNA is nearly absent outside of the nucleolus. This statement is very surprised considering the author's own results and all known published papers on the topic.
- The functional experiment performed in figure 4 is interesting, however there is no quantification of the results obtained. This is a general problem in the whole manuscript. Data are shown as one showcase, without not even indication of the n (number of nuclei imaged), number of replicates and

some basic quantifications with statistics. Although the data show some potentially interesting findings they are very preliminary and cannot be trusted if some reproducibility is not ensured.

- The authors show underwound DNA to be in proximity with RNA Pol II then they focus on the nucleolus and RNA Pol I. The two sections of the paper seem to be quite unrelated, no connection is made.

-

Reviewer #2 (Remarks to the Author):

The authors present a study on in-situ detection of negative supercoiled DNA using psoralen crosslinking. They correlated the resulting labeling with markers for the three main components of nucleoli, FC, DFC, and GC in untreated cells and those incubated with inhibitors of rRNA transcription and nucleophosmin binding to G4 quadruplexes. The authors conclude that the binding of underwound DNA in the GC restricts axial rotation of rDNA induced by rRNA transcription in the fibrillar components of nucleoli. Therefore, the authors suggest that the binding of nucleophosmin to G4 structures in the affects the positioning of rRNA transcription within the fibrillar components.

The results in the presented paper are of great interest in the field and would warrant publication in Communications Biology. To my opinion, there are some points that need to be addressed by the authors:

(a) The authors have performed many control experiments. However, I miss one experiment that would massively underline the validity of the method and at the same time would further support the drawn conclusion, and this is the correlation of psoralen staining with detection of rDNA by FISH. FISH under conditions for rDNA detection would definitely prove that psoralen detects DNA. Furthermore, FISH under conditions to detect rRNA (if a probe from the transcription unit is used) can act as a more general marker of GC than NPM. I strongly recommend to add this experiment.

(b) 2.3. §5,10: the statements made here are not well supported by Fig. 3C, which is an essential figure in the paper. To underline the authors' statement, it would require some kind of measurement, e.g. line scans and/or scatterplots. In contrast to the authors, I see no clear colocalization between psoralen and any of the 3 markers (see also previous point). I would like to point out that if my interpretation holds true it could further underline the authors' conclusions.

(c) Throughout the paper there are statements and interpretations that seem to be somewhat contradictory. In Discussion, §5 the authors correctly state here that G4 may preferentially form in promoters of rRNA genes. Beforehand, the authors also correctly state that rRNA transcription takes place in the fibrillar components of nucleoli and throughout the paper they propose that underwound rDNA is constrained in the GC (see also Fig. 4B!). The authors should comment on this putative contradiction.

Minor points:

(d) CX incubation using 3µM for 3 hours appears to be a pretty harsh treatment. So far, mostly milder treatments with CX were published in studies using other cells. Cells have different sensitivities to CX exposure but for several cell types this would be a lethal dose. Have the authors checked if the cells are not simply dying after 3 hours?

(e) To my opinion, certain aspects are missing (mostly in Discussion): (A) what other factors might add to constraints of underwound rDNA within nucleoli (proteins, rRNA); (B) mentioning of the enormous differences in torque/strain generation between RNAPI and RNAPII-driven transcription; (C) rRNA genes differ in their transcriptional activity that likely has an influence on the extent of underwound rDNA.

(f) P4 §5: Here, the authors state that transcription takes place in the FC while in Fig. 4B transcription takes place at the interface between FC and DFC. Please clarify as there are differing views in the field.

Reviewer #3 (Remarks to the Author):

Fukute and collaborators evaluate the levels of negative supercoiling in the nucleus using super-resolution imaging in combination with incorporation of biotinylated psoralen into the DNA. With this technique, the authors propose that negative supercoiling accumulates in the nucleoli due to RNAPII transcription and that such phenomenon depends on the anchoring of DNA to the shell of this nuclear condensate. Although this is a compelling technique and the results could be interesting in the field, several controls are needed to conclude that the shell of the nucleoli is the anchor of DNA in terms of supercoiling.

1. In Fig. 1, it would be convenient the use of topoisomerase inhibitors (camptothecin, merbarone, ICRF-193) to show that changes in supercoiling can be detected with the method, not just bleomycin that induces a lot of DNA breaks and reduces dramatically psoralen incorporation.
2. In Fig. 4, the bio-psoralen signal completely disappears after the treatment with TMPyP4. I wonder whether there is still DNA in this region under the treatment, DNA FISH experiments could proof that rDNA is still maintained in the structure, besides the reduced biopsoralen incorporation.
3. In Fig. 4, the authors need to proof that the binding of ribosomal genes to NPM1 is affected by the treatment with TMPyP4. ChIP experiments might be an option. Is it possible to disrupt the shell with a different approach? Maybe downregulation of NPM1. In fact, NPM1 signal is reduced under the treatment with TMPyP4. This should be explained.

Reviewer #1 (Remarks to the Author):

Fukute et al. performed some experiments to study the function of the nucleolus in the constrain of DNA axial rotation for DNA underwinding. The data are potentially interesting but very preliminary. Some suggestions follow:

We are grateful to the reviewer for the encouraging and constructive comments that have helped us improve the manuscript. All comments have been carefully considered and appropriate changes have been made in the revised manuscript. The response to each individual comment is given in blue font.

(a) The authors should mention the existency of overwound DNA in the introduction and results. Making just a distinction between underwound and relaxed DNA this can bias the interpretation of the results.

We agree with the reviewer's comment. Indeed, the manuscript should mention the possibility of the existence of overwound DNA both in the Introduction and Results. Especially, as discussed in the response to the reviewer's comment (e), we have made changes to some of the statements related to overwound DNA.

We amended the corresponding parts in **Introduction (Page (P.) 2, Line (L.) 2–15)** and **Results (P. 3, L. 31-33)**.

(b) Section 2.1: "Biotinylated psoralen binds to underwound DNA in the nucleus" this has been previously shown by numerous publications cited in the manuscript and others. Just to make an example ref 22 (Naughton et al NSMB 2013) has mapped the presence of underwound and overwound regions in human cells and have stained nuclei with bTMP showing how bleomycin treatment decreases bTMP signal. As this is previous knowledge it cannot be presented as original findings.

We thank the reviewer for pointing to this aspect. Naughton et al. succeeded with fluorescence imaging using biotinylated trimethylpsoralen (bTMP) and examined its specificity using bleomycin. In our study, we used (non-trimethylated) biotinylated psoralen (bio-psoralen), for which specific binding to underwound DNA has not been validated by fluorescent imaging. Therefore, we needed to confirm this specific binding with Bleomycin treatment, according to the method used by Naughton.

We have amended the corresponding part in **Results (P. 3, L. 19-21)**, to clarify the purpose of the experiments by directly discussing Naughton's study.

(c) Supplementary figure 1 is quite unclear. What are we supposed to see in the agarose gel between the different samples and in heated/not heated samples? Is the signal measured in the plot in C the agarose band intensity? Also, to be able to compare signals between control, TopI and Gyrase conditions (as done in S1C) it would be appropriate to show all bands from a single gel. Can the authors provide this?

We apologize for the unclear description of the figure in the original version. As the reviewer has understood, we compared the input (non-heated) and heated samples by measuring the respective band intensities. When the bio-psoralen is intercalated into and crosslinks DNA by UV exposure, the crosslinked DNA is protected from heat denaturation. As shown in the modified Supp. Fig. 1b, dsDNA was detected on a single gel for all samples (without/with heat denaturation, for each condition [Control, Top I and Gyrase]). The band intensity in the heated sample relative to that of the input sample for each condition was calculated.

We have amended **Supp. Fig. 1**.

(d) The control carried out to exclude binding of psoralen to RNA (Supp Fig 2B) is not clear. It would be better to perform a quantification to exclude co-localization. Psoralen seems to be localized in spots in the nucleus, can you explain why it has this type of distribution?

We thank the reviewer for pointing out this important issue, considering that ribosome RNA is produced in the nucleolus, where prominent signals of psoralen are found in this study. The psoralen signal was diminished by DNase I treatment (the new Supp. Fig. 2b), supporting the finding that psoralen is bound to DNAs. In addition, Lattice-SIM² microscopy showed distinct distributions of psoralen and RNA within the nucleolus, as shown in a line scan (the new Supp. Fig. 2c). In addition, RNase A treatment decreased the RNA signal, whereas psoralen signals were retained (the new Supp. Fig. 2d-f). Collectively, these results support that psoralen is preferentially bound to DNA in this study.

We have added **Supp. Fig. 2b-f**.

(e) Figure 2. The authors detect a large amount of Bio-psoralen outside the nucleus. This is quite surprising. They conclude that this is due underwound DNA in mitochondria claiming colocalization with TFAM. It is not clear how much of the signal colocalize with mitochondrial markers as the images of bio-psoralen and TFAM do not show a very clear colocalization pattern and a quantification of the extent of colocalization is not provided. Also, the picture shown (2B, it would be nice to see more images) displays a very large amount of extranuclear signal, even more intense than the nuclear signal. It is important to ensure that the process of permeabilization with digitonin/Bio-psoralen incorporation/UV irradiation has not created

some leakage of nuclear DNA. Moreover, in 2C DAPI and psoralen signal exclude each other. Does DAPI not stain underwound DNA? Also, authors say “DAPI stained DNA in a relaxed state” disregarding the existence of overwound DNA. In Fig 2D the authors refer to bio-psoralen positive regions as underwound DNA but the presence of DNA marked by DAPI is not clear in these regions. In Fig 2E, according to normal staining of Top1 one would expect Top1 signal a more extended area of the nucleus and not exclusively in few bio-psoralen foci. How do the authors explain this?

We appreciate the reviewer's important comments. The protocol especially utilizing digitonin needed careful discussion about the possible leakage of DNAs/proteins. We have corrected the overstatement on cytoplasmic signals in the Results and have added the **Discussion (P. 6, L. 35 - P. 7, L. 5)**, to describe technical limitation in our method. Nevertheless, to ensure our main claim, we obtained fundamental data indicating that the “core-shell” nucleolar structure with ribosome DNA and nucleolar proteins is retained (**the new Fig. 3a, 5e and the new Supp. Fig. 4**). Detailed responses are given below.

- We apologize for any confusion related to the previous version of Fig. 2b (TFAM staining). To avoid from any misleading/over-statement, we have **removed the previous version of Fig. 2b** and corresponding sentences from the Results.

- It was also misleading to state that “DAPI stained DNA in a relaxed state”. In the study by Escandarte et al. (*Sci. Rep.* 2016), it has been rather suggested that DAPI forms stronger intermolecular interactions with overwound DNA, which has a decreased minor groove width. We amended the corresponding parts in **Results (P. 3, L. 31-33)** with additional references (Refs. 35 and 36).

35. Estandarte, A. K., Botchway, S., Lynch, C., Yusuf, M. & Robinson, I. The use of DAPI fluorescence lifetime imaging for investigating chromatin condensation in human chromosomes. *Sci. Rep.* **6**, 31417 (2016).

36. Suto, R. K. et al. Crystal structures of nucleosome core particles in complex with minor groove DNA-binding ligands. *J. Mol. Biol.* **326**, 371–380 (2003).

- As the reviewer commented, Top1 would span a more extended area (nucleoplasm) as reported by Girstun et al. (2017). Although we have checked that ribosome DNA and nucleolar proteins are retained in the nucleolus in this study, it is possible that some portion of proteins leak to cytoplasm by digitonin treatment.

We have added a sentence to mention this important point and have amended the corresponding part in **Results (P. 4, L. 5-7)**.

(f) The authors stated “we minimized the digitonin treatment time to 1 min and confirmed that the core–shell structure was conserved after bio-psoralen imaging.” There are no results included for this important control. Indeed, as I mentioned above digitonin treatment might cause trivial leakage of nuclear DNA in the cytoplasm.

We thank the reviewer for pointing out this important issue. We have performed lattice-SIM for nucleolar proteins in control and digitonin-treated cells (**the new Supp. Fig. 4**). Based on this result, we confirmed that component proteins are retained in the nucleolus.

We have amended the corresponding parts in **Results (P. 4, L. 26-30)** and have added a new paragraph mentioning this technical limitation in **Discussion (P. 6, L. 35 - P. 7, L. 5)**.

(g) Part of section 2.3 is based on “as RNAPI inhibition simultaneously disrupts the nucleolar structure that likely is necessary for DNA underwinding” implying that the nucleolar proteins rather than transcription itself are needed for underwinding. This is a bit misleading as it might imply that RNAPI is dispensable for underwinding or not the main driver, authors should consider rephrasing this.

We thank the reviewer for this suggestion. We have now changed this statement to “Having identified the driving force of DNA underwinding in the nucleolus, we next explored possible factors that constrain DNA axial rotation.” in **Results (P. 5, L. 10-11)**.

(h) The authors show that underwound DNA is generated in nucleolar FC and DFC. This is interesting, however they also stated that underwound DNA is nearly absent outside of the nucleolus. This statement is very surprising considering the author’s own results and all known published papers on the topic.

We apologize for the misleading statement in the original version of the manuscript. As shown in Fig. 2b, underwound DNA is also distributed outside the nucleolus where RNA Pol II generates the driving force. We have now rephrased the statement “nearly absent outside the nucleolus” to “less abundant outside the nucleolus” in **Results (P. 4, L. 35)**.

(i) The functional experiment performed in figure 4 is interesting, however there is no quantification of the results obtained. This is a general problem in the whole manuscript. Data are shown as one showcase, without not even indication of the n (number of nuclei imaged), number of replicates and some basic quantifications with statistics. Although the data show some potentially interesting finding they are very preliminary and cannot be trusted if some reproducibility is not ensured.

We thank the reviewer for this important comment. For the data in the previous Fig. 4a, the fluorescence intensities of bio-psoralen inside and outside the NPM signal were

quantified and compared in **the new Fig. 5c and d**. Also, in the figures listed below, statistical tests were performed with *n* indicating the number of samples and repetitions.

Fig. 3e: Mean intensity of bio-psoralen inside and outside the nucleolus

Fig. 4d: Mean intensity of bio-psoralen inside and outside the nucleolus in CX-5461- and ActD-treated cells.

Fig. 5a: Normalized percentage of input calculated in ChIP-qPCR

Fig. 5d: Mean intensity of bio-psoralen inside and outside the nucleolus in TMPyP4-treated cells

Supp. Fig. 2e: Fluorescence intensity of RNA upon RNase A treatment

Supp. Fig. 2f: Fluorescence intensity of bio-psoralen upon RNase A treatment

Supp. Fig. 3b: Fluorescence intensity of bio-psoralen upon TopI/II inhibitor treatment

(j) The authors show underwound DNA to be in proximity with RNA Pol II then they focus on the nucleolus and RNA Pol I. The two sections of the paper seem to be quite unrelated, no connection is made.

We understand the reviewer's concern. In this study, the observation of underwound DNA in proximity with RNA Pol II, which was in agreement with the fact that RNA Pol II generates torque to induce DNA underwinding, was one of the bottom lines to consider the methodology to be functional.

To explain the relevance better, we have amended the corresponding parts in **Results (P. 3, L. 36- P. 4, L. 1)**, to clarify the purpose of RNA Pol II staining.

Reviewer #2 (Remarks to the Author):

The authors present a study on in-situ detection of negative supercoiled DNA using psoralen crosslinking. They correlated the resulting labeling with markers for the three main components of nucleoli, FC, DFC, and GC in untreated cells and those incubated with inhibitors of rRNA transcription and nucleophosmin binding to G4 quadruplexes. The authors conclude that the binding of underwound DNA in the GC restricts axial rotation of rDNA induced by rRNA transcription in the fibrillar components of nucleoli. Therefore, the authors suggest that the binding of nucleophosmin to G4 structures in the affects the positioning of rRNA transcription within the fibrillar components.

The results in the presented paper are of great interest in the field and would warrant publication in *Communications Biology*. To my opinion, there some points that need to be addressed by the authors:

We thank to the reviewer for the positive assessment of the relevance of our study and suggestions for the improvement of the manuscript. The response to individual comments is given in blue font.

(a) The authors have performed many control experiments. However, I miss one experiment that would massively underline the validity of the method and at same time would further support the drawn conclusion, and this is the correlation of psoralen staining with detection of rDNA by FISH. FISH under conditions for rDNA detection would definitely prove that psoralen detects DNA. Furthermore, FISH under condition to detect rRNA (if a probe from the transcription unit is used) can act as more general marker of GC than NPM. I strongly recommend to add this experiment.

We thank the reviewer for suggesting this important control experiment. Based on this suggestion, we performed FISH for rDNA and confirmed that bio-psoralen signal overlapped with that of rDNA (**the new Fig. 3a**). In addition, rDNA signal was sustained after TMPyP4 treatment, whereas psoralen signals were reduced (**the new Fig. 5e**). These results directly demonstrate that, indeed, rDNA is underwound in the nucleolus. Unfortunately, the probe functioned for rDNA FISH did not target 5'ETS (Ref. 42) and therefore the probe could not be used as a GC marker. Nevertheless, we believe that this experiment directly detecting rDNA has greatly strengthened the manuscript.

42. Wen, Y. R. et al. Nascent pre-rRNA sorting via phase separation drives the assembly of dense fibrillar components in the human nucleolus. *Mol. Cell* **76**, 767-783.e11 (2019).

(b) 2.3. §5,10: the statements made here are not well supported by Fig. 3C, which is an essential figure in the paper. To underline the authors statement, it would require some kind of measurement, e.g. line scans and/or scatterplots. In contrast to the authors, I see no clear colocalization between psoralen and any of the 3 markers (see also previous point). I would like to point out that if my interpretation holds true it could further underline the authors conclusions.

We appreciate the reviewer for raising another important suggestion to improve the manuscript. We have added a line scan analysis in **Fig. 3c** and the quantification of bio-psoralen signal inside/outside the nucleolar shell in **the new Fig. 3d and e**.

(c) Throughout the paper there are statements and interpretations that seem to be somewhat contradictory. In Discussion, §5 the authors correctly state here that G4 may preferentially form in promoters of rRNA genes. Beforehand, the authors also correctly state that rRNA transcription takes place in the fibrillar components of nucleoli and throughout the paper they propose that underwound rDNA is constrained in the GC (see also Fig. 4B!). The authors should comment on this putative contradiction.

We apologize for any confusion caused by misleading texts and figures. In the previous version of Fig. 4b, the promoter was placed on the right and the coding region on the left. It looked as if G4 was formed in the coding region. In the revised figure (**the new Fig. 5f**), the direction of DNA movement has been reversed, where the promoter is on the left and the coding region is on the right. In this figure, RNAP transcribes rRNA while reeling rDNA from right to left. In the GC, the axial rotation of DNA is constrained by NPM–G4 interaction, contributing to DNA underwinding in FC and DFC (promoter regions of rDNA). We believe that the revised figure correctly explains our conclusion.

(d) CX incubation using 3 μ M for 3 hours appears to be a pretty harsh treatment. So far, mostly milder treatments with CX were published in studies using other cells. Cells have different sensitivities to CX exposure but for several cell types this would be a lethal dose. Have the authors checked if the cells are not simply dying after 3 hours?

According to the reviewer's suggestion, we performed dead cell staining at different time points (0 h [no treatment], 15 min, 3 h, and 24 h) of CX-5461 treatment. As the reviewer had indicated, we observed a time-dependent increase in the percentage of dead cells (**the new Supp. Figure 5**). To make sure that our observation is relevant, we repeated the psoralen staining with the treatment time of 15 min, and confirmed that psoralen foci is diminished (**the new Fig. 4**). We have also included the result using Actinomycin D, as another control for RNAP I inhibition in the new Fig. 4.

(e) To my opinion, certain aspects are missing (mostly in Discussion): (A) what other factors might add to constraints of underwound rDNA within nucleoli (proteins, rRNA); (B) mentioning of the enormous differences in torque/strain generation between RNAPI and RNAPII-driven transcription; (C) rRNA genes differ in their transcriptional activity that likely has an influence on the extent of underwound rDNA.

We greatly appreciate the suggestion of the reviewer regarding important aspects to be discussed. We have added following points in **Discussion (P. 6, L. 17-34)**.

(A) There are other possible factors that constrain DNA axial rotation. For example, DNA looping by upstream binding factors (UBF) and G4 formation itself could be constraints.

(B) We cited papers on the kinetics of RNAP where authors showed that RNAP I transcribes faster and pauses less often than RNAPII. This finding suggested that RNAP I has a stronger torque capacity than RNAPII.

(C) We agree that variation in transcriptional activities of rDNA repeats affect the amount of underwound DNA. The non-uniform distribution of underwound DNA along rDNA may reflect differences in the transcriptional activity of each rDNA repeat.

(f) P4 §5: Here, the authors state that transcription takes place in the FC while in Fig. 4B transcription takes place at the interface between FC and DFC. Please clarify as there are differing views in the field.

We thank the reviewer for this observation. A super resolution analysis showed that nascent rRNA was extruded from the surface of FC (Wen et al., *Mol. Cell.* 76, 767–783.e11, 2019), suggesting that transcription occurs between FC and DFC. Based on this report, we have revised the statement to “transcription takes place at the FC/DFC interface” in **Results (P. 4, L. 25)**.

Reviewer #3 (Remarks to the Author):

Fukute and collaborators evaluate the levels of negative supercoiling in the nucleus using super-resolution imaging in combination with incorporation of biotinylated psoralen into the DNA. With this technique, the authors propose that negative supercoiling accumulates in the nucleoli due to RNAPII transcription and that such phenomenon depends on the anchoring of DNA to the shell of this nuclear condensate. Although this is a compelling technique and the results could be interesting in the field, several controls are needed to conclude that the shell of the nucleoli is the anchor of DNA in terms of supercoiling.

We greatly appreciate the positive comments and expert suggestions. We believe that the comments helped us to further strengthen our manuscript, by performing experiments with additional inhibitors, rDNA-FISH, and ChIP-qPCR. The response to individual comments is given in blue text.

(a) In Fig. 1, it would be convenient the use of topoisomerase inhibitors (camptothecin, merbarone, ICRF-193) to show that changes in supercoiling can be detected with the method, not just bleomycin that induces a lot of DNA breaks and reduces dramatically psoralen incorporation.

As important control experiments, we have added the results of treatment with the TopI and TopII inhibitors, camptothecin and etoposide, respectively, to **the new Supp. Fig. 3**. Contrary to our expectations, Top I/II inhibition reduced fluorescence intensity of bio-psoralen in the nucleus. Naughton et al. (2013) made a similar observation that Top I/II inhibition leads to decrease in psoralen-incorporation at transcription start sites. As discussed in the Results section, the side effects of Top I/II inhibitor treatment would decrease underwound DNA by introducing DNA strand breaks as well as impairing transcriptional activities.

We have added the explanation above in **Results (P. 4, L. 7-13)**.

(b) In Fig. 4, the bio-psoralen signal completely disappears after the treatment with TMPyP4. I wonder whether there is still DNA in this region under the treatment, DNA FISH experiments could proof that rDNA is still maintained in the structure, besides the reduced biopsoralen incorporation.

We thank the reviewer for this very important comment. We performed FISH for rDNA and confirmed that bio-psoralen signal overlapped with that of rDNA (**the new Fig. 3a**). In addition, rDNA signal was sustained inside the nucleolus after TmPyP4 treatment, whereas psoralen signal was reduced (**the new Fig. 5e**). These results demonstrate that rDNA is underwound in the nucleolus, and is relaxed by TMPyP4 treatment.

(c) In Fig. 4, the authors need to prove that the binding of ribosomal genes to NPM1 is affected by the treatment with TMPyP4. ChIP experiments might be an option. Is it possible to disrupt the shell with a different approach? Maybe downregulation of NPM1. In fact, NPM1 signal is reduced under the treatment with TMPyP4. This should be explained

We appreciate the reviewer for providing the idea of control experiments on NPM binding to G4. Based on your suggestion, we have performed a ChIP experiment followed by quantitative PCR (**the new Fig. 5a**) and confirmed that NPM–G4 interaction was inhibited by TMPyP4 treatment. We believe that the result directly supports the main claim of this manuscript.

We also tried downregulation of NPM by two different Silencer[®] Select siRNAs (ThermoFisher Scientific) targeting different coding regions; however, we were not successful, probably because the siRNAs are not validated by the manufacturer.

As the reviewer pointed out, our result showed that TMPyP4 treatment reduces NPM signal in the nucleolus. In a previous paper by Chiarella et al. (2013), NPM is reported to diffuse into the nucleoplasm upon TMPyP4 treatment. We have added the above explanation to the **Results (P. 5, L. 20-22)**.

REVIEWERS' COMMENTS:

Reviewer #1 (Remarks to the Author):

The authors addressed the reviewer's questions and the manuscript has significantly improved. I have no major comment, just the following remark:

- the authors say "Although we expected that TopI/II inhibition would increase underwound DNA and produce a stronger bio-psoralen signal, Top I/II inhibition decreased the nuclear bio-psoralen signal, indicating a decrease in the amount of underwound DNA. Such responses, as previously reported²³, might be caused by secondary effects of TopI/II inhibitor treatment, such as DNA strand break⁴⁰ and reduced RNAP activity⁴¹." The decreased bTMP intensity upon Top inhibition was previously reported in reference 21, which can be incorporated in this section.

Reviewer #2 (Remarks to the Author):

In their revised manuscript, the authors go to great lengths to address my previous points of criticism. I would particularly like to emphasize the FISH experiments, which in my opinion significantly substantiate the manuscript. All other points were also well addressed, so I can recommend publication in its present form.

Reviewer #3 (Remarks to the Author):

The revised manuscript has substantially improved, and most of my concerns have been addressed. Only minor points should be corrected:

-In Figure 5e, the authors state that rDNA remains in the nucleoli after TMPyP4 treatment. However, mock control is missing. The image of that control should be added to the figure.

-In figure legend of supplementary figure 3. Camptoesin should be corrected by Camptothecin.

-In page 7, line 227, "whereas excess DNA underwinding is known to stall RNAP in the promoter region" should be corrected since it is the opposite effect. It should be changed to: "whereas excess DNA underwinding is known to promote the release of RNAP from the promoter region".

Reviewer #1 (Remarks to the Author):

The authors addressed the reviewer's questions and the manuscript has significantly improved. I have no major comment, just the following remark:

- the authors say "Although we expected that TopI/II inhibition would increase underwound DNA and produce a stronger bio-psoralen signal, Top I/II inhibition decreased the nuclear bio-psoralen signal, indicating a decrease in the amount of underwound DNA. Such responses, as previously reported²³, might be caused by secondary effects of TopI/II inhibitor treatment, such as DNA strand break⁴⁰ and reduced RNAP activity⁴¹." The decreased bTMP intensity upon Top inhibition was previously reported in reference 21, which can be incorporated in this section.

We are grateful to the reviewer for the encouraging comments. We incorporated the reference 21 in the corresponding part.

Reviewer #2 (Remarks to the Author):

In their revised manuscript, the authors go to great lengths to address my previous points of criticism. I would particularly like to emphasize the FISH experiments, which in my opinion significantly substantiate the manuscript. All other points were also well addressed, so I can recommend publication in its present form.

We thank to the reviewer for helping us to strengthen the manuscript and for the recommendation for the manuscript.

Reviewer #3 (Remarks to the Author):

The revised manuscript has substantially improved, and most of my concerns have been addressed. Only minor points should be corrected:

We thank to the reviewer for further assessment. We corrected the manuscript as below.

(a) In Figure 5e, the authors state that rDNA remains in the nucleoli after TMPyP4 treatment. However, mock control is missing. The image of that control should be added to the figure.

Thank you for the important advice. We included the result of Mock control in the Fig. 5e.

(b) In figure legend of supplementary figure 3. Camptoesin should be corrected by Camptothecin.

We apologize for the Typo. We have corrected the spelling to "Camptothecin".

(c) In page 7, line 227, "whereas excess DNA underwinding is known to stall RNAP in the promoter region" should be corrected since it is the opposite effect. It should be changed to:

“whereas excess DNA underwinding is known to promote the release of RNAP from the promoter region”.

We thank to the reviewer for the professional comment. We have changed the expression to “whereas excess DNA underwinding is known to promote the release of RNAP from the promoter region”.